# Incidence of Mortality, Acute Kidney Injury and Graft Loss in Adult Kidney Transplant Recipients with Coronavirus Disease 2019: Systematic Review and Meta-Analysis

**DOI:** 10.3390/jcm10215162

**Published:** 2021-11-04

**Authors:** Jia-Jin Chen, George Kuo, Tao Han Lee, Huang-Yu Yang, Hsin Hsu Wu, Kun-Hua Tu, Ya-Chung Tian

**Affiliations:** 1Department of Nephrology, Chang Gung Memorial Hospital, Linkou Main Branch, Taoyuan City 33305, Taiwan; raymond110234@hotmail.com (J.-J.C.); b92401107@gmail.com (G.K.); kate0327@hotmail.com (T.H.L.); hyyang01@gmail.com (H.-Y.Y.); tomwu38@gmail.com (H.H.W.); christopher.tu@gmail.com (K.-H.T.); 2Department of Nephrology, Kidney Research Center, Chang Gung Memorial Hospital, Taoyuan City 33305, Taiwan

**Keywords:** acute kidney injury, coronavirus disease 2019, kidney replacement therapy, graft loss, mortality

## Abstract

The adverse impact of Coronavirus disease 2019 (COVID-19) on kidney function has been reported since the global pandemic. The burden of COVID-19 on kidney transplant recipients, however, has not been systematically analyzed. A systematic review and meta-analysis with a random-effect model was conducted to explore the rate of mortality, intensive care unit admission, invasive mechanical ventilation, acute kidney injury, kidney replacement therapy and graft loss in the adult kidney transplant population with COVID-19. Sensitivity analysis, subgroup analysis and meta-regression were also performed. Results: we demonstrated a pooled mortality rate of 21% (95% CI: 19−23%), an intensive care unit admission rate of 26% (95% CI: 22–31%), an invasive ventilation rate among those who required intensive care unit care of 72% (95% CI: 62–81%), an acute kidney injury rate of 44% (95% CI: 39–49%), a kidney replacement therapy rate of 12% (95% CI: 9–15%), and a graft loss rate of 8% (95% CI: 5–15%) in kidney transplant recipients with COVID-19. The meta-regression indicated that advancing age is associated with higher mortality; every increase in age by 10 years was associated with an increased mortality rate of 3.7%. Regional differences in outcome were also detected. Further studies focused on treatments and risk factor identification are needed.

## 1. Introduction

The novel coronavirus, severe acute respiratory syndrome coronavirus 2 (SARS-CoV-2), caused COVID-19 from the end of 2019 and has resulted in a huge burden on global healthcare systems. As of June 2021, more than 170 million people had been infected and nearly 4 million died. COVID-19 consists of a primary pulmonary infection with extensive systemic involvement. The overwhelming inflammatory response may lead to cytokine storm and multi-organ failure.

In severe COVID-19, dysregulated immunity induces endothelial injury, complement-mediated thrombosis and microangiopathy. The kidneys are one of the organs most involved during the progress of the disease; therefore, acute kidney injury (AKI) is common in patients with COVID-19 [1]. The incidence of AKI in COVID-19 varies across populations and critically ill patients seem to be the most susceptible [2,3]. In addition to being one of the negative impacts of COVID-19, AKI also serves as a predictor of mortality in patients with COVID-19 [4]. The kidney function reserve varies from patient to patient, depending on the kidneys’ ability to handle external stresses or hazardous stimuli. In patients undergoing cardiac surgery, poorer preoperative kidney function is associated with a higher possibility of postoperative AKI [5]. Compared with the general population, kidney transplant recipients have a lower average kidney function reserve; thus, they are more susceptible to AKI [6,7].

Kidney transplant recipients are constantly at risk of complications associated with immunosuppression, which include opportunistic infections (e.g., BK virus, Epstein–Barr virus and cytomegalovirus infections), post-transplant lymphoproliferative disorder, and complications associated with immunosuppressants (e.g., calcineurin inhibitor-associated nephrotoxicity, calcineurin inhibitor- and corticosteroid-associated new-onset diabetes after transplantation and dyslipidemia) [8]. Because of their immunosuppressed status, the kidney transplant population is more susceptible to infection than the general population. The risk of COVID-19 transmission from a household contact is also higher in patients with solid organ or stem cell transplant [9]. In several studies, the reported outcomes of COVID-19 were worse in kidney transplant recipients with COVID-19 compared with the general population [10,11], although this has not been systematically analyzed. The aim of this study was to systematically review and analyze the outcomes of COVID-19 in kidney transplant recipients, including mortality rate, acute kidney injury rate, invasive ventilation rate and rate of graft loss.

## 2. Materials and Methods

### 2.1. Literature Search Strategy

We performed this meta-analysis in accordance with the preferred reporting items for systematic reviews and meta-analyses (PRISMA) guidelines for a meta-analysis of observational studies (Appendix A) [12]. We registered the protocol in PROSPERO (CRD42021260803). Two independent reviewers (J.J. Chen and G. Kuo) comprehensively searched for studies published before 08 June 2021 on PubMed, Medline, the Cochrane Library and Embase. The search strategy targeted published clinical trials, cohort studies, case series, letters to the editor and commentaries. The keyword and Mesh term used on PubMed were: (((COVID-19) OR (SARS-CoV-2)) OR (coronavirus)) AND ((((Kidney Transplantation) OR (Kidney transplant)) OR (Renal transplantation)) OR (Renal transplant)) with the following filters: Humans, Adult: 19+ years. English-language articles that were published from 2019–2021 were screened.

The other detailed search strategy and the results of that search process are provided in Appendix A. Review articles and meta-analyses were not included in our analysis, but their references were screened and searched for relevant studies.

### 2.2. Study Eligibility Criteria

The titles and abstracts of the studies returned by the search were examined by two reviewers (J.J. Chen and G. Kuo) independently, and articles were excluded upon initial screening of their titles or abstracts if these indicated that they were clearly irrelevant to the objective of the current study. The full texts of relevant articles were reviewed to determine whether the studies were eligible for inclusion. The inclusion criteria are: (i) studies enrolled an adult population with confirmed COVID-19, (ii) studies enrolled kidney transplant recipients, (iii) studies reported at least one of the outcomes of interest. The third reviewer (T.H. Lee) was consulted to reach an agreement through consensus in the case of any disagreement regarding eligibility. Studies were excluded if they were duplicated cohorts, presented insufficient information of outcomes, or included a pediatric population or a population of more than one organ transplantation.

### 2.3. Data Extraction and Outcome

Two investigators (J.J. Chen and G. Kuo) independently extracted the outcomes of interest and the characteristics of the included studies. The primary outcome in the present study was mortality in adult kidney transplant patients with COVID-19. The secondary outcomes included AKI, kidney replacement therapy (KRT), ICU admission, invasive mechanical ventilation (IMV), and graft loss. The IMV rate was calculated by the number of patients receiving IMV divided by the number of patients admitted to the ICU. The graft loss rate was calculated by the number of patients with graft loss divided by the number of patients who survived after COVID-19 infection.

### 2.4. Data Synthesis and Analysis

The analysis was conducted using the metaprop function in R package meta (version 4.18-2) [13]. The event rates of the outcomes of interest, including the mortality, ICU admission, AKI, IMV, KRT, and graft loss, were pooled and estimated. We chose the random-effect model because it is methodologically conservative for summary estimates and is more suitable as potential heterogeneity in the study populations may exist. Heterogeneity was examined by *I*^2^ (≥50% indicates substantial heterogeneity) and Cochran’s Q statistic (*p* < 0.1 indicates moderate heterogeneity) [14,15,16]. The *p* values were two-sided, and statistical significance was set at *p* < 0.05. The sensitivity analysis was performed by removing the studies with a moderate-to-high risk of bias. The subgroup analysis was performed to explore potential sources of heterogeneity according to the study design (retrospective or prospective), location (single center or multi-center), patient number (<30 or at least 30), level of care (mixed in-patient and out-patient or purely in-patient), and areas of the countries where the study conducted (North America, Latin America, Europe, Asia). In the meta-regression, age and sample size were considered as potential mortality modifiers. Publication bias was assessed by the funnel plot and Egger’s test; a *p* value of the Egger’s test of < 0.1 indicates potential publication bias [16]. The statistical analysis was performed by using R software version 4.1.0 (The R Foundation, Vienna, Austria) [17].

### 2.5. Risk of Bias Assessment

The quality of the cohort studies was assessed independently by two authors (J.J. Chen and G. Kuo) using the Newcastle–Ottawa scale [18], which allocates a maximum of nine points for three major domains: quality of the selection, comparability, and outcome of study populations. Studies with a score of 7–9 were considered as low risk of bias, those with a score of 4–6 were considered as moderate risk of bias and those with a score of less than 4 were considered as high risk of bias. Disagreements between the two investigators (J.J. Chen and G. Kuo) were resolved by consensus with another author (T.H. Lee).

## 3. Results

### 3.1. Study Characteristics

The literature search flow is shown in Appendix A. Through the electronic database search, there were 371 potentially eligible studies from PubMed, 496 potentially eligible studies from EMBASE, 385 studies from Medline, and 0 studies from the Cochrane review. After removing the duplicated articles, a total of 750 articles were screened according to their titles and abstracts. Seventy-eight full-text articles were further assessed for eligibility after screening (Appendix A). After excluding 19 studies for various reasons, 59 articles were included (Appendix A). A total of 59 studies comprising 5956 participants were enrolled in our study. Most of the studies were retrospective in design and enrolled hospitalized kidney transplant recipients with COVID-19. A minority of studies included a mixed level of care, where kidney transplant recipients with COVID-19 were managed either in the hospital or in an outpatient clinic (Table 1).

### 3.2. Mortality of Adult Transplant Population with COVID-19 Infection

We found 58 studies with 5948 patients that reported the mortality of COVID-19 in adult kidney transplant recipients. The pooled mortality rate calculated by the random effect model was 21% (95% confidence interval [CI]: 19−23%) with moderate heterogeneity (*I*^2^ = 57%, *p* < 0.01) (Figure 1).

### 3.3. Secondary Outcomes of Adult Transplant Population with COVID-19 Infection

Thirty-two studies reported the rate of ICU admission. The pooled ICU admission rate was 26% (95% CI: 22−31%) with moderate heterogeneity (*I*^2^ = 80%, *p* < 0.01). The rates of IMV among patients requiring ICU care were reported in 21 studies. The pooled IMV rate was 72% (95% CI: 62–81%) with moderate heterogeneity (*I*^2^ = 65%, *p* < 0.01). The incidence rate of AKI was reported in 38 studies, with a pooled rate of 44% (95% CI: 39−49%) and moderate heterogeneity (*I*^2^ = 61%, *p* < 0.01). The requirement of KRT among patients experiencing AKI was reported in 27 studies. The pooled KRT rate among AKI patients was 30% (95% 22−39%) with high heterogeneity (*I*^2^ = 83%, *p* < 0.01), and was 12% (9−15%) with high heterogeneity (*I*^2^ = 83%, *p* < 0.01) among the whole-kidney transplant population. The rate of graft loss was reported in only 11 studies. The pooled graft loss rate among kidney transplant COVID-19 survivors was 8% (95% CI: 5−15%) with a high heterogeneity (*I*^2^ = 87%, *p* < 0.01). The summarized and detailed information of these secondary outcomes is depicted in Figure 2.

### 3.4. Publication and Risk of Bias of Enrolled Studies

Potential publication bias was illustrated by a funnel plot (Appendix A). The Egger’s test of funnel asymmetry displayed an insignificant result (*p* value = 0.64). The risk of bias was assessed via the Newcastle–Ottawa scale for non-randomized studies; we summarize the risk of bias of each study in Appendix A. For retrospective studies with a single-center population, we considered that there was a potential risk regarding the representativeness of the exposed cohort. Owing to the retrospective study design in nature, most of the enrolled studies were without control group and therefore the domain score regarding the comparability of cohorts was considered as zero.

### 3.5. Sensitivity Analysis, Subgroup Analysis and Meta-Regression

The sensitivity analysis was performed by including only studies with a low risk of bias (NOS scale higher or equal to 7). A total of eight studies fulfilled this criterion. The pooled mortality was 20% (95% CI: 18−22%) with low heterogeneity (*I*^2^ = 35%, *p* = 0.15). (Figure 3A).

We performed a subgroup analysis by dividing the studies into different groups: (1) study design, (2) single- or multi-center, (3) patient number, (4) level of care, and (5) areas of study countries. There were no interactions for study design, single or multi-center, patient numbers, or level of care. There was significant interaction between the different areas of study, with a higher mortality rate reported for Latin America. (*p* < 0.01) (Figure 3B).

A meta-regression was performed to examine the relationship between two covariates (sample size and age) and mortality. Sample size was not significantly associated with increased mortality risk (Figure 4A). The mortality rate was significantly associated with older age (mortality rate = 0.0037 × Age − 0.0092, *p* value = 0.023). This implies a 3.7% increase in mortality rate for every 10 years of age (Figure 4B).

## 4. Discussion

In the present systematic review and meta-analysis, two points are worth noting. First, we demonstrated higher rates of adverse outcomes in adult kidney transplant recipients with COVID-19, and these included mortality (21%), ICU admission (26%), IMV among those who required ICU care (72%), AKI (44%), KRT (12%) and graft loss (8%). Second, older age is associated with an increase in mortality rate.

Compared to the general population, the incidence of AKI and KRT and the mortality rate in kidney transplant recipients are higher [22,24,78,79]. In the non-transplant population, Meyerowitz-Katz et al. reported a pooled infection fatality rate of 0.68% among COVID-19 patients [78]. Levin et al. reported that the fatality rate was increased with advanced age. The age-specific fatality rate of COVID-19 increased from less than 0.4% below age 55 to 15% at age 85 [80]. In kidney transplant recipients, the excessive deaths may reflect their immunocompromised status and susceptibility to infection. Using the registry data of Australia and New Zealand, Chan et al. demonstrated a higher infection-related mortality rate in kidney transplant recipients compared to the general population. Ozturk et al. compared the clinical characteristics of COVID-19 among kidney transplant recipients, HD, CKD and control groups (patients without kidney diseases) [65]. Overall, the kidney transplant recipients demonstrated a higher mortality risk than the control group, but were still less vulnerable to death than CKD and HD patients [65]. In addition to the severity of COVID-19 per se, co-infection with other viruses can increase the risk of disease progression and mortality in kidney transplant recipients. Molaei et al. reported that co-infection of COVID-19 and cytomegalovirus, the most prevalent pathogen in kidney transplant recipients, might increase mortality risk [60]. In addition, potential drug–drug interactions between immunosuppressive medications and antiviral therapy could complicate and worsen the clinical condition [81]. In patients who recovered from COVID-19, those with immunocompromised status may still experience longer viral shedding from the respiratory tract than the general population [82]. The increased susceptibility to infection and delayed viral clearance both indicate that immunocompromised status contributes to higher mortality in kidney transplant recipients. These findings may explain the higher mortality risk among kidney transplant recipients when compared to the general population.

In the present study, we demonstrated that older age is a risk factor of mortality in kidney transplant recipients with COVID-19. The observation of age-related mortality is in agreement with the study by Chan et al. [83]. This is not surprising, because age has been shown to be a crucial risk factor for mortality in different populations [46,84,85,86].

AKI is common during the routine care of kidney transplant recipients. In a retrospective longitudinal cohort study using the US Renal Data System, 11.6% of kidney transplant recipients experienced episodes of AKI, and 14.8% of those who experienced AKI required temporary KRT [6]. This higher susceptibility to AKI is also observed in COVID-19 patients. According to the meta-analysis published by Chen et al., the pooled AKI occurrence in the general population was 8.9% [79]. Among the studies enrolled in the meta-analysis by Chen et al., there were two studies reporting the AKI rates of kidney transplant recipients with COVID-19, which were 30–57% [22,24]. The use of calcineurin inhibitors by kidney transplant recipients can cause vascular and endothelial damage. In COVID-19, vessels and endothelium are targets for viral attack. The baseline susceptibility to endothelial injury by an external insult to the vasculature may make kidney transplant recipients more vulnerable to microvascular injury and thrombosis [87,88,89]. In our study, the pooled AKI rate was 43% in the kidney transplant recipients, which is strikingly higher than in the general population. Among those patients with AKI, 30% required KRT. The KRT rate was higher than in previous reports on the general population, and this indicates that COVID-19 might pose a higher risk of AKI and severe AKI requiring dialysis in kidney transplant recipients than in the general population.

In this study, the rate of ICU admission (26%) and IMV rate (73%) were similar to the data from the general population. In a meta-analysis, Chang et al. included 12,437 ICU-admitted COVID-19 patients, 69% of whom required IMV [90]. However, the actual rate of ICU admission and mechanical ventilation may be difficult to compare directly between studies because the disease prevalence may vary between different times and countries. The capacity of ICUs and their criteria of admission may also differ among healthcare systems.

In the present study, we also observed a difference in mortality across different areas. Another meta-analysis demonstrated that the mortality of COVID-19 patients with at least one comorbidity is higher in Europe and Latin America [91]. This observation, however, should be interpreted with caution. The differences between healthcare systems and the timing and variants of the SARS-CoV-2 pandemic, the implementation of non-pharmaceutical intervention against viral spreading, and the speed of vaccination could lead to variability in the extent and severity of COVID-19. The relationship between mortality and geographic region requires further observation and investigation through larger-scale, multinational studies.

This study features some limitations. First, information on the detailed immunosuppressive regimens, the concentration of immunosuppression medications, the type of the transplants (deceased or living) and the induction therapy regimens are lacking in most studies. Therefore, we could not adjust the outcomes with these factors. Among the studies with smaller patient numbers, the majority of baseline immunosuppressive agents were similar to those used in our daily practice, which included a combination of prednisolone, mycophenolate mofetil/mycophenolic acid, and tacrolimus. A smaller fraction of patients receive cyclosporine instead of tacrolimus, and a much smaller proportion of patients take leflunomide or azathioprine. The adjustment of immunosuppressants took the form of either a reduction in the dosage or temporary discontinuation. Basuki et al. reported a lower mortality with cyclosporine in kidney transplant recipients than with other immunosuppressive drugs during the treatment of COVID-19 [92]. However, only small and limited studies reported the differences between immunosuppressive regimens. Larger registries contain no detailed data for the comparison. Second, most studies do not report the severity of patients’ symptoms by any known critical care scoring systems; therefore, we could not classify patients’ disease severity beyond dividing them into an inpatient population and a population of patients managed on an outpatient basis. In addition, advancements in COVID-19 knowledge and management might result in improvements in COVID-19 prognosis. However, owing to this, information on COVID-19 treatment strategies is lacking in most studies. We analyzed the possible impact of earlier and recent recruitment of patients on clinical outcomes using subgroup analysis and meta-regression by stratifying the studies according to the date of final assessment or enrollment before or after July 2020. Using these two analysis models, we did not observe any significant difference in mortality between early and recent studies (data not shown). Furthermore, most studies are retrospective in design and based on a single -enter population. Although we performed a meta-regression examining the relationship between sample size and mortality, the risk of bias regarding the lack of representativeness and adequate comparability should be noted.

## 5. Conclusions

Our study demonstrated that adult kidney transplant recipients with COVID-19 had a high rate of mortality, AKI, and KRT. The risk of mortality increased in proportion with the recipients’ age. Further studies focused on risk factor stratification, immunosuppressant drugs adjustment, and antiviral treatment in kidney transplant populations with COVID-19 infection are needed.

## Figures and Tables

**Figure 1 jcm-10-05162-f001:**
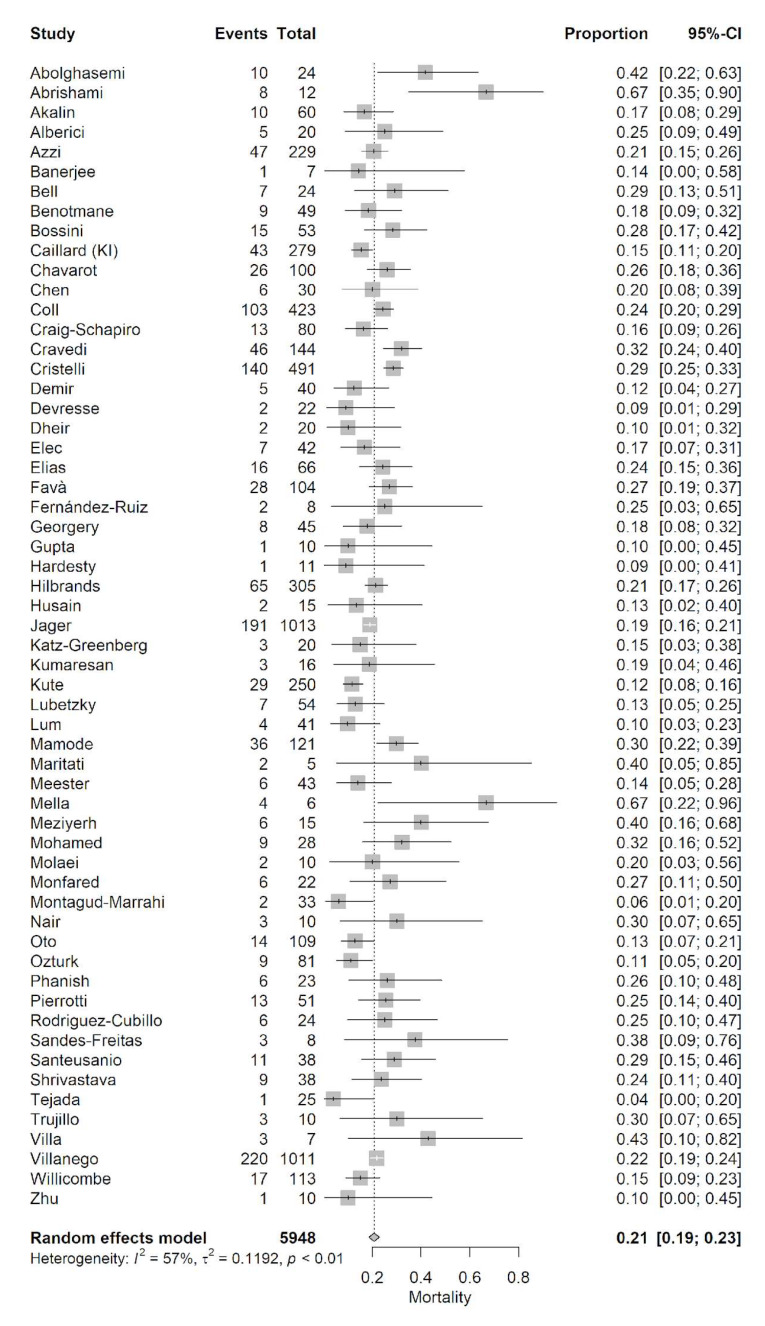
Forest plot of pooled incidence of mortality in the adult kidney transplant population with COVID-19 infection.

**Figure 2 jcm-10-05162-f002:**
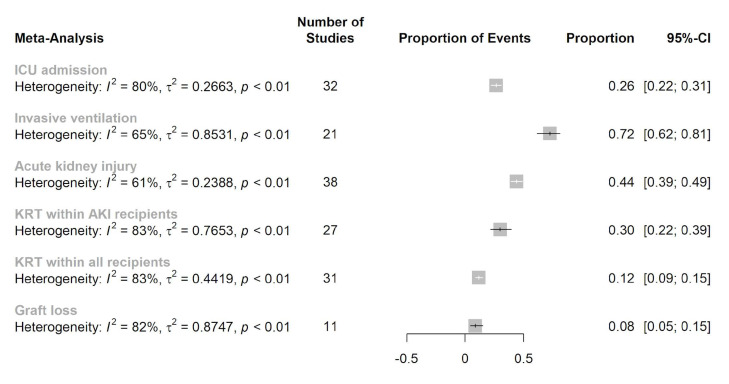
Forest plot of secondary outcome in the adult kidney transplant population with COVID-19 infection.

**Figure 3 jcm-10-05162-f003:**
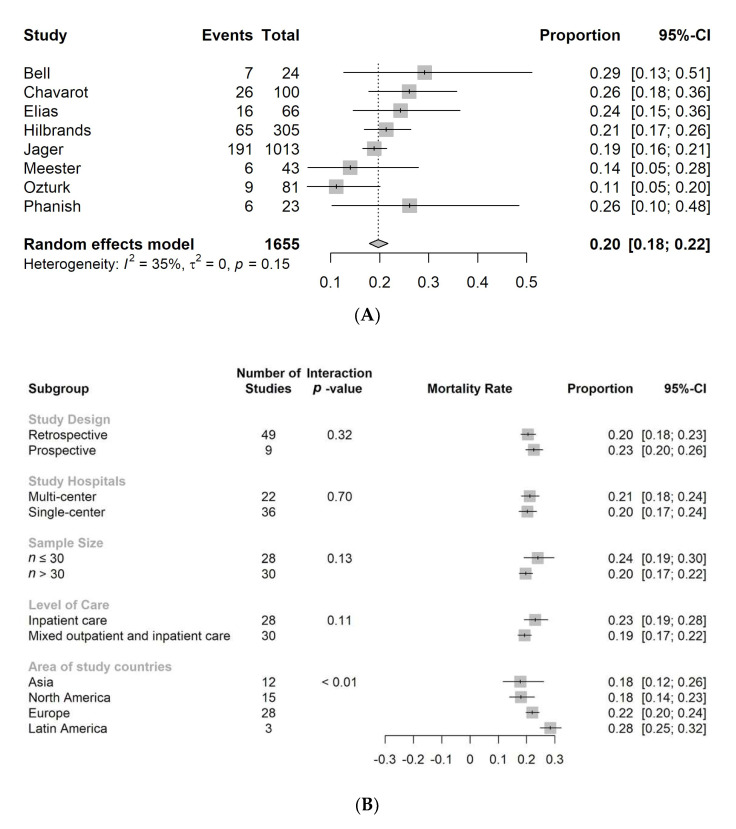
Sensitivity analysis after excluding studies with moderate-to-high risk of bias ((**A**), upper) and forest plot of subgroup analysis ((**B**), lower).

**Figure 4 jcm-10-05162-f004:**
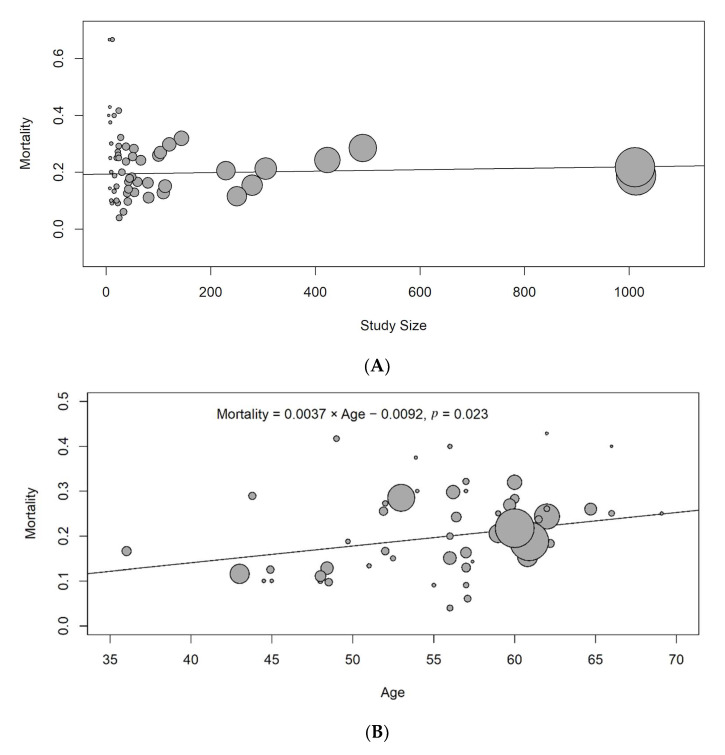
Meta-regression according to sample size ((**A**), upper) and age ((**B**), lower).

**Table 1 jcm-10-05162-t001:** Characteristics and outcomes of the included studies.

Study	Design	Country	Location	NOS	Age (Mean, y/o)	Sample Size	Outcome					
							Mortality	ICU	MV	AKI	KRT	Graft Loss
Abolghasemi [19]	RC	Iran	Multi-center	6	49	24	10	12	NR	NR	NR	NR
Abrishami [20]	RC	Iran	Single-center	5	47.6	12	8	10	9	NR	NR	NR
Akalin [21]	RC	USA	Single-center	5	36	60	10	NR	11	NR	6	NR
Alberici [22]	RC	Italy	Single-center	5	59	20	5	4	0	6	1	NR
Azzi [23]	PC	USA	Single-center	5	59	229	47	NR	NR	NR	NR	NR
Azzi [23]	PC	USA	Single-center	5	61	79 ^##^	NR	NR	28	18	18	5
Banerjee [24]	RC	UK	Multi-center	6	57.4	7	1	4	2	4	3	NR
Bell [25]	RC	Scotland (Scottish Renal Registry)	Multi-center	7	NR	24	7	NR	NR	NR	NR	NR
Benotmane [26]	RC	France	Single-center	5	62.2	49	9	14	2	31	4	NR
Bossini [27]	RC	Italy	Multi-center	6	60	53	15	10	9	15	3	NR
Caillard [28]	RC	France (nationwide registry)	Multi-center	6	60.8	279	43	88	72	106	27	9
Chavarot [29]	RC	France	Multi-center	7	64.7	100	26	34	29	NR	NR	NR
Chen [30]	RC	USA	Single-center	5	56	30	6	NR	7	NR	4	NR
Coll [31]	RC	Spain	Multi-center	6	62	423	103	57	36	NR	NR	NR
Craig-Schapiro [32]	RC	USA, NY	Single-center	6	57	80	13	NR	16	25	4	4
Cravedi [33]	RC	USA	Multi-center	6	60	144	46	NR	42	74	NR	NR
Cristelli [34]	PC	Brazil	Single-center	5	53	491	140	NR	156	229	155	NR
Demir [35]	RC	Turkey	Multi-center	6	44.9	40	5	7	6	14	NR	NR
Devresse [36]	PC	USA	Single-center	5	57	22	2	2	2	5	0	NR
Dheir [37]	RC	Turkey	Single-center	5	48	20	2	NR	2	5	2	NR
Elec [38]	RC	Romania	Single-center	5	52	42	7	8	NR	10	N	NR
Elhadedy [39]	RC	UK	Multi-center	6	50.1	8	0	1	1	2	1	0
Elias [40]	PC	France	Multi-center	7	56.4	66	16	15	15	28	7	NR
Favà [41]	RC	Spain	Multi-center	6	59.7	104	28	NR	14	47	NR	NR
Fernández-Ruiz [42]	RC	Spain	Single-center	5	69.1	8	2	0	0	NR	NR	NR
Georgery [43]	RC	Belgium	Single-center	5	60.6	45	8	14	14	NR	NR	NR
Gupta [44]	PC	India	Single-center	5	44.5	10	1	NR	NR	NR	NR	NR
Hardesty [45]	RC	USA	Single-center	6	55	11	1	NR	3	1	1	NR
Hilbrands [46]	PC	Europe, ERACODA collaboration	Multi-center	7	60	305	65	57	49	30+	30	NR
Husain [47]	RC	USA, NY (Columbia University Vagelos College of Physicians and Surgeons)	Single-center	5	51	15	2	NR	4	6	NR	NR
Jager [48]	PC	Europe, ERA-EDTA Registry	Multi-center	7	60.9	1013	191	NR	NR	NR	NR	NR
Katz-Greenberg [49]	RC	USA, Philadelphia	Single-center	5	52.5	20	3	NR	4	9	1	1
Kumaresan [50]	RC	India	Single-center	5	49.7	16	3	NR	3	1	1	NR
Kute [51]	RC	India	Multi-center	6	43	250	29	53	30	121	24	12
Lubetzky [52]	RC	USA	Single-center	5	57	54	7	NR	11	21	3	3
Lum [53]	RC	USA	Single-center	5	48.5	41	4	9	8	11	4	1
Mamode [54]	RC	UK	Multi-center	5	56.2	121	36	30	22	NR	19	15
Maritati [55]	RC	Italy	Single-center	5	66	5	2	NR	3	1	1	NR
Meester [56]	PC	Belgium (NBVN Kidney Registry Group)	Multi-center	7	NR	43	6	NR	NR	NR	NR	NR
Mella [57]	RC	Italy	NR	5	55.5	6	4	NR	2	NR	NR	NR
Meziyerh [58]	RC	Netherlands	Single-center	5	56	15	6	6	5	NR	NR	NR
Mohamed [59]	PC	UK	Single-center	6	57	28	9	5	NR	14	2	NR
Molaei [60]	RC	Iran	Single-center	5	59.6	10	2	4	4	7	NR	NR
Monfared [61]	RC	Iran	Single-center	5	52	22	6	NR	5	12	NR	NR
Montagud-Marrahi [62]	RC	Spain	Single-center	5	57.1	33	2	13	2	NR	NR	1
Nair [63]	RC	USA, NY	Single-center	5	57	10	3	5	4	5	1	NR
Oto [64]	RC	Turkey	Multi-center	6	48.4	109	14	22	19	46	4	NR
Ozturk [65]	RC	Turkey	Multi-center	7	48	81	9	17	14	NR	NR	NR
Phanish [66]	RC	UK	Multi-center	7	62	23	6	9	6	13	4	2
Pierrotti [67]	RC	Brazil	Single-center	5	51.9	51	13	23	17	30	19	0
Rodriguez-Cubillo [68]	RC	Spain	Single-center	5	66	24	6	NR	5	14	3	0
Sandes-Freitas [69]	RC	Brazil (National registry)	Multi-center	6	53.9	8	3	4	4	4	4	2
Santeusanio [70]	RC	USA	Single-center	5	43.8	38	11	NR	14	22	12	12
Shrivastava [71]	RC	USA	Single-center	5	61.5	38	9	13	9	27	6	NR
Tejada [72]	RC	USA, Detroit Medical Center	Single-center	5	56	25	1	4	1	16	NR	NR
Trujillo [73]	RC	Spain	Single-center	5	54	10	3	NR	0	8	NR	NR
Villa [74]	RC	Germany	Single-center	6	62	7	3	NR	3	4	NR	NR
Villanego [75]	PC	Spain (prospectively filled registry)	Multi-center	6	60	1011	220	140	NR	NR	NR	NR
Willicombe [76]	RC + PC ^#^	UK, London	Single-center	5	56	113	17	NR	NR	NR	NR	NR
Zhu [77]	RC	China	Multi-center	6	45	10	1	NR	0	6	0	NR

^#^ Including a seroprevalence survey and clinical cohort; ^##^ Sub-cohort report different outcomes other than mortality; abbreviations: AKI, Acute kidney injury; ICU, intensive care unit; KRT, kidney replacement therapy; MV, mechanical ventilation, NOS, Newcastle–Ottawa Scale; NR: not reported; PC, prospective cohort/case series; RC, retrospective cohort/case series.

## Data Availability

Not applicable.

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
