# Peer review of "Incidence of Mortality, Acute Kidney Injury and Graft Loss in Adult Kidney Transplant Recipients with Coronavirus Disease 2019: Systematic Review and Meta-Analysis"

_jcm, 2021, doi:10.3390/jcm10215162_

Round 1

Reviewer 1 Report

I read this systematic review with some interest. However, some issues must be addressed before publication can be advised and the writing needs close attention to the grammar and vocabulary. 

Specific comments:

  1. The abstract should be a maximum of 200 words as per the journal's guidelines to authors.
  2. Please align all terminologies with the forthcoming KDIGO nomenclature guidelines. The word ‘renal’ should be replaced with ‘kidney’.
  3. "... has been reported after the global pandemic" - isn't the pandemic still ongoing?
  4. Please change "coronavirus disease 2019" to "Coronavirus Disease 2019 (COVID-19)".
  5. Please note that "SARS-CoV-2" is short for "Severe acute respiratory syndrome coronavirus 2".
  6. "... incidence of COVID-19 in the kidney transplant population might be similar to that in the general population" - immunosuppressed people are more likely to have a breakthrough infection (even if they have been vaccinated) and are also more likely to transmit the virus to household contacts (see: pubmed.ncbi.nlm.nih.gov/33185244).
  7. Please change "which are not yet been systematic discussed" to "which has not been systematically analysed". Please also ensure consistency of tenses throughout the manuscript.
  8. In the main manuscript, suggest authors provide the full electronic search strategy used to identify studies, including all search terms and limits for at least one database. The rest can be displayed in the supplementary materials.
  9. Why was the P value of the Egger’s test set at < 0.1? At least some explanation is required.
  10. What was the level of significance for the other analyses?
  11. Some comments about the study quality and risk of bias assessment should be included in the discussion section.
  12. Kidney transplant recipients are constantly at risk of complications associated with immunosuppression, namely, opportunistic infections (e.g., Epstein-Barr virus and cytomegalovirus infections), post-transplant lymphoproliferative disorder, and complications associated with immunosuppressants (e.g., calcineurin inhibitor- and corticosteroid-associated new-onset diabetes after transplantation and calcineurin inhibitor-associated nephrotoxicity). This should be appropriately mentioned and cited (citation: pubmed.ncbi.nlm.nih.gov/33712863).
  13. What were the common immunosuppressants/regimen used in the different studies? At least some comments would be relevant.

Author Response

Reviewer 1

Comments and Suggestions for Authors I read this systematic review with some interest. However, some issues must be addressed before publication can be advised and the writing needs close attention to the grammar and vocabulary.

Point 1. The abstract should be a maximum of 200 words as per the journal's guidelines to authors.

Reply 1: Thanks  for your kindly reminding us. We have checked the total word count count of our abstract and it is within the limitation of the journal's suggestion. The final word count of the abstract in the revised manuscript is 184 .

Point 2. Please align all terminologies with the forthcoming KDIGO nomenclature guidelines. The word ‘renal’ should be replaced with ‘kidney’.

Reply 2: Thanks  for your kindly reminding us. We replaced the renal transplantation with kidney transplantation and renal replacement therapy with kidney replacement therapy throughout the revised manuscript.

Point 3. "... has been reported after the global pandemic" - isn't the pandemic still ongoing?

Reply 3: Thanks for your opinion, we have modified our sentence into “….has been reported after since the global pandemic.”

Point 4. Please change "coronavirus disease 2019" to "Coronavirus Disease 2019 (COVID-19)".

Reply 4: Thanks  for your kindly reminding us. We have changed “coronavirus disease 2019” into “Coronavirus Disease 2019”  or “COVID-19" in the revised manuscript.

Point 5. Please note that "SARS-CoV-2" is short for "Severe acute respiratory syndrome coronavirus 2".

Reply 5: Thanks  for your kindly reminding us. We have added the full name "Severe acute respiratory syndrome coronavirus 2" into the revised manuscript where "SARS-CoV-2" was first used.

Point 6. "... incidence of COVID-19 in the kidney transplant population might be similar to that in the general population" - immunosuppressed people are more likely to have a breakthrough infection (even if they have been vaccinated) and are also more likely to transmit the virus to household contacts (see: pubmed.ncbi.nlm.nih.gov/33185244).

Reply 6: Thanks for your insightful comment, we have added this description according to your viewpoint into revised manuscript in introduction session with above mentioned article (tracking change version page 2, line 54)

Point 7. Please change "which are not yet been systematic discussed" to "which has not been systematically analysed". Please also ensure consistency of tenses throughout the manuscript.

Reply 7: Thanks for your valuable opinion. According to your suggestion, we have changed "which are not yet been systematic discussed" into "which has not been systematically analysed" in the revised manuscript.

Point 8. In the main manuscript, suggest authors provide the full electronic search strategy used to identify studies, including all search terms and limits for at least one database. The rest can be displayed in the supplementary materials.

Reply 8: Thanks for your valuable opinion. We have added the detail search strategy we used in pubmed into the main manuscript according to your suggestion. (tracking change version page 2, line 69)

Point 9. Why was the P value of the Egger’s test set at < 0.1? At least some explanation is required.

Reply 9: Thanks for your insightful comment. The reported cutoff P value of Egger’s test for detection publication bias in previous published studies are 0.05, 0.1 and even to 0.2. The initial cutoff value chosen by Sterne JA and Egger M was 0.1 (reference 1).  Hayashino and his colleagues demonstrated that with increased cut-off P value, the sensitivity could increase without a substantial increment of false positive rate (reference 2). Recently published review by Lin et. al. and Song et. al.  also suggested a P value of 0.1 as cutoff point owing to the statistical power of the publication bias tests is generally low (reference 3, 4).

Reference 1: Sterne JA, Egger M. Funnel plots for detecting bias in meta-analysis: guidelines on choice of axis. J Clin Epidemiol. 2001 Oct;54(10):1046-55. doi: 10.1016/s0895-4356(01)00377-8. PMID: 11576817.

Reference 2: Hayashino Y, Noguchi Y, Fukui T. Systematic evaluation and comparison of statistical tests for publication bias. J Epidemiol. 2005 Nov;15(6):235-43. doi: 10.2188/jea.15.235. PMID: 16276033; PMCID: PMC7904376.

Reference 3: Lin L, Chu H, Murad MH, Hong C, Qu Z, Cole SR, Chen Y. Empirical Comparison of Publication Bias Tests in Meta-Analysis. J Gen Intern Med. 2018 Aug;33(8):1260-1267. doi: 10.1007/s11606-018-4425-7. Epub 2018 Apr 16. PMID: 29663281; PMCID: PMC6082203.

Reference 4: Song F, Khan KS, Dinnes J, Sutton AJ. Asymmetric funnel plots and publication bias in meta-analyses of diagnostic accuracy. Int J Epidemiol. 2002 Feb;31(1):88-95. doi: 10.1093/ije/31.1.88. PMID: 11914301.

Point 10. What was the level of significance for the other analyses?

Reply 10: Thanks for your important opinion. We have added this sentence “ P values were 2-sided, and statistical significance was set at P < .05. “ into the method session in the revised manuscript. (tracking change version page 3 line 105)

Point 11. Some comments about the study quality and risk of bias assessment should be included in the discussion section.

Reply 11: Thanks for your kindly reminding us. We have added a small session about study quality and risk of bias in the result and discussion session (tracking change version page 8, line 173; page 12, line 292)

Point 12. Kidney transplant recipients are constantly at risk of complications associated with immunosuppression, namely, opportunistic infections (e.g., Epstein-Barr virus and cytomegalovirus infections), post-transplant lymphoproliferative disorder, and complications associated with immunosuppressants (e.g., calcineurin inhibitor- and corticosteroid-associated new-onset diabetes after transplantation and calcineurin inhibitor-associated nephrotoxicity). This should be appropriately mentioned and cited (citation: pubmed.ncbi.nlm.nih.gov/33712863).

Reply 12: Thanks for your insight comment and kindly suggestion. We have added this important background information into the introduction session in the revised manuscript, and added a new citation. (tracking change version page 2, line 47)

Point 13. What were the common immunosuppressants/regimen used in the different studies? At least some comments would be relevant.

Reply 13: Thanks for your comment, we have addressed this issue more detailedly in the discussion session. Most of the large studies did not precisely show the complete baseline immunosuppressive regimen as well as how the immunosuppressive drugs were adjusted. In small studies, the baseline immunosuppressive agents were similar to our daily practice, which mostly included a combination of prednisolone, mycophenolate mofetil/mycophenolic acid, and tacrolimus. A smaller fraction of patients receive cyclosporine instead of tacrolimus, and a much smaller proportion of patients take leflunomide or azathioprine. Most of the studies did not show the vintage after transplantation in each patient, the regimens or doses of the immunosuppressive agents were difficult to compare between each other.  (tracking change version page 12, line 273)

Reviewer 2 Report

In this systematic review and meta-analysis, Chen et al. explore the incidence of mortality, acute kidney injury and graft loss in adult kidney transplant recipients with COVID-19.

Particularly, the authors clearly demonstrate mortality rates of 21% (95% CI:19%-23%), intensive care unit admission rate of 26% (95% CI: 22-31%), invasive ventilation rate among those who require intensive care unit care of 72% (95% CI: 62-81%), 19 acute kidney injury rate of 44% (95% CI: 39-49%), kidney replacement therapy of 12% (95% CI: 9-15%) 20 and graft loss of 8% (95% CI: 5-15%) in kidney transplant recipients with coronavirus disease 2019.

A meta-analysis indicated that higher age is associated with a mortality, with regional differences in outcomes.

In summary, this is a comprehensive and timely study important to understand effects of this pandemic disease in patients susceptible to severe disease course, including kidney transplant recipients.

I recommend acceptance in its current version. 

Author Response

Thanks for your reviewing and kindly comment. We are glad to hear your suggestion about acceptance of our manuscript. Thanks for your effort in reviewing our article.

Round 2

Reviewer 1 Report

Thank you for the revisions.